# Role of Nanomedicine-Based Therapeutics in the Treatment of CNS Disorders

**DOI:** 10.3390/molecules28031283

**Published:** 2023-01-28

**Authors:** Zi-Hua Guo, Saadullah Khattak, Mohd Ahmar Rauf, Mohammad Azam Ansari, Mohammad N. Alomary, Sufyan Razak, Chang-Yong Yang, Dong-Dong Wu, Xin-Ying Ji

**Affiliations:** 1Department of Neurology, Kaifeng Hospital of Traditional Chinese Medicine, No. 54 East Caizhengting St., Kaifeng 475000, China; 2Henan International Joint Laboratory for Nuclear Protein Regulation, School of Basic Medical Sciences, Henan University, Kaifeng 475004, China; 3Department of Surgery, Miller School of Medicine, University of Miami, Miami, FL 33136, USA; 4Henan-Macquarie University Joint Centre for Biomedical Innovation, School of Life Sciences, Henan University, Kaifeng 475004, China; 5Department of Epidemic Disease Research, Institute for Research & Medical Consultations (IRMC), Imam Abdulrahman Bin Faisal University, P.O. Box 1982, Dammam 31441, Saudi Arabia; 6National Centre for Biotechnology, King Abdulaziz City for Science and Technology (KACST), P.O. Box 6086, Riyadh 11442, Saudi Arabia; 7Dow Medical College, John Hopkins Medical Center, School of Medicine, Baltimore, MD 21205, USA; 8School of Nursing and Health, Henan University, Kaifeng 475004, China; 9School of Stomatology, Henan University, Kaifeng 475004, China

**Keywords:** central nervous system disorders, blood–brain barrier, nanomedicine, immunotherapy, nanotechnology

## Abstract

Central nervous system disorders, especially neurodegenerative diseases, are a public health priority and demand a strong scientific response. Various therapy procedures have been used in the past, but their therapeutic value has been insufficient. The blood–brain barrier (BBB) and the blood–cerebrospinal fluid barrier is two of the barriers that protect the central nervous system (CNS), but are the main barriers to medicine delivery into the CNS for treating CNS disorders, such as brain tumors, Parkinson’s disease, Alzheimer’s disease, and Huntington’s disease. Nanotechnology-based medicinal approaches deliver valuable cargos targeting molecular and cellular processes with greater safety, efficacy, and specificity than traditional approaches. CNS diseases include a wide range of brain ailments connected to short- and long-term disability. They affect millions of people worldwide and are anticipated to become more common in the coming years. Nanotechnology-based brain therapy could solve the BBB problem. This review analyzes nanomedicine’s role in medication delivery; immunotherapy, chemotherapy, and gene therapy are combined with nanomedicines to treat CNS disorders. We also evaluated nanotechnology-based approaches for CNS disease amelioration, with the intention of stimulating the immune system by delivering medications across the BBB.

## 1. Introduction

Nanomedicine is a new field that utilizes nano-scale resources for applications in the diagnosis and treatment of diseases [1]. Several nano-scale materials are used in nanomedicine: organic, inorganic, carbon-based, polymeric, liposomes, extracellular vesicles, and metals [2,3,4]. Boron-containing compounds (BCCs) have a wide range of medicinal effects [5,6,7]. Currently, only a few BCCs, such as vaborbactam, tavaborole, crisaborole, and bortezomib, are permitted for use in people. However, many other BCCs are demonstrating promising results as prospective therapies for human disorders, particularly neurological conditions. Consequently, the interest in BCCs on models of neuronal disorders is rising. These include studies that reveal beneficial activity on multiple targets, those that demonstrate a pro-survival impact on human-derived SH-SY5Y cells, such as an A-toxicity model, and those that demonstrate ameliorative abilities in both in vitro and in vivo models of AD [8,9,10,11]. The relationship between nanoscience and pharmaceutical science is broad and there are emerging applications in various fields of disease diagnosis and therapeutics [12,13,14]. Tumor cells are passively targeted by nanomedicines through enhanced retention and permeability (EPR) [15]. The improved permeability and retention are used to target the lymphatic and microvascular systems in the tumor’s interior [16,17,18]. Alternatively, nanomedicine approaches in active and functional targeting are widely examined to overcome particular challenges in tumor theragnostic [19].

The term “central nervous system (CNS) disorders” refers to a broad range of significant neurological conditions, the majority of which have no effective treatments [20]. Gliomas and glioblastomas, two types of brain cancer, are inherited diseases that originate from cells known as neuroglial ancestor cells [21]. Glioblastoma, which makes up 15% of primary brain tumors and 50% of all gliomas, is the most typical primary CNS tumor in adults. Only temozolomide (TMZ) and the antiangiogenic medicine bevacizumab have been approved by the US Food and Drug Administration (FDA) in recent years for the treatment of gliomas [22]. The median survival rate for people with glioblastoma is less than 2 years despite recent treatment advancements [23]. The more common neurodegenerative disorders Parkinson’s disease (PD) and Alzheimer’s disease (AD) affect millions of people worldwide [24]. There are several drugs for AD that have been licensed by the FDA, such as cholinesterase inhibitors (e.g., rivastigmine and donepezil) as well as NMDA receptor antagonists (e.g., memantine), However, they have limited benefits on severe cognitive impairment and cannot reverse the course of the condition [25,26,27]. The only drug approved to treat Parkinson’s disease is Xadago (safinamide) that reduces motor symptoms without addressing the underlying pathophysiology of the disease [28,29]. Preventing disease development and treating the symptoms and pathology after a late-stage diagnosis are the major challenges in this profession. A new wave of treatment strategies has emerged as a result of these challenges. Currently, radiation, chemotherapy, gene therapy, immunotherapy, and surgery are used to treat CNS illnesses; however, every therapeutic approach has advantages and disadvantages [30,31]. Immunotherapy uses the immune system of the host to target disease cells by enhancing or suppressing innate immune responses [32]. Active immunotherapy and passive immunotherapy are the two categories of immunotherapeutic techniques [33]. Nanomedicines, tumor vaccines, and non-specific immune stimulants are all examples of active immunotherapeutic techniques that aim to elicit an immune response [34,35,36]. Through administering lymphocytes or antibodies to patients, passive immunotherapy promotes anticancer effects [37]. Therefore, while implementing immunotherapeutic techniques, it is important to take into account the distinct immunological milieu of the CNS [38,39]. The presence of complement components, the expression of Toll-like receptors, and the existence of microglia, astrocytes, and pericytes as antigen-presenting cells (APCs) demonstrate that the CNS immune system exists [40]. Even with a disrupted blood-brain barrier (BBB), certain invading cancers contain no enhancing areas, which restricts therapeutic medication interaction despite the immune system’s dynamic and adaptive characteristics [41]. Additionally, the delivery of medicines across the BBB is a substantial problem due to the tremendous adaptive characteristics of glioblastoma, in addition to its comparatively low immunogenicity, development of an inhibiting tumor microenvironment (TME), and intertumoral heterogeneity. In fact, many scientists have proposed that the CNS may indeed be regarded as an “immunologically inactive” location, providing a distinct and balanced environment that favors the predominance of immunosuppressive mediator production [42].

The fundamental challenge that restricts the efficacy of immunotherapies is the engagement of surrogate immunosuppression by brain tumors via several pathways. Overexpressed markers (CD4+, CD25+, and FOXP3+) on regulatory T cells (Tregs) in the sick state influence the immunologically cold TME [43]. Glioblastomas are a particular kind of immunoprivileged tumor since they are sometimes referred to as “cold TMEs” and are only peripherally penetrated by immune cells. A cold TME may also be influenced by immune-suppressive cytokines like interleukins IL-6 and IL-10, as well as immune-suppressive cytokines like transforming growth factor-beta (TGF-β) [44]. PD-1 (programmed cell death protein 1) is overexpressed by inactive Tregs, which aids tumors in evading CNS immune responses and results in a cold TME [45,46]. As a result, a number of tightly controlled checkpoints maintain brain immunity. Immunotherapy has undergone several changes as a result of training the immune system to detect disease locations [47]. One is immune checkpoint inhibition, in which medications (typically antibodies) block immunological checkpoints that tumors have overexpressed in order to reveal cancer cells and ultimately activate immune responses against the tumors (e.g., melanoma) [48]. Alternatively, immune responses can be triggered by genomically altered targeted therapeutics (such as chimeric antigen receptor T cells (CAR T cells)), which have been altered to detect and cure the patient’s malignancy [49]. Immunotherapy for the treatment of CNS illnesses continues to be a significant therapeutic issue despite decades of scientific study. For instance, immunological checkpoints and CAR T-cell therapy are less effective in treating nonresponsive cancers that have fewer mutations and neoantigens [50]. Immunotherapy can be particularly difficult due to the location and morphologic similarity of nonmalignant cells with neuroglial cells [51]. A significant barrier to obtaining a better therapeutic effect for CNS illnesses is the difficulty in deliver therapeutically appropriate dosages to the disease region.

Drug clearance via the kidneys, drug circulation time in the blood, medication penetration through the blood–brain barrier (BBB) and blood–brain tumor barrier (BBTB) are further obstacles to drug delivery into the central nervous system (CNS) [52]. The payload’s ability to enter the CNS is frequently constrained by a natural defense mechanism of efflux pumps, such as multidrug-resistant protein [53] and convection-enhanced diffusion [54]. In order to overcome these difficulties, CNS treatments are being developed using an active and functionally focused nanomedicine-based strategy [55]. The emerging field of nanomedicine uses nanoscale materials for a wide range of purposes in diseases diagnosis and treatment [1]. The BBB needs to be crossed by effective nanomedicines for CNS illnesses, and different parameters need to be tuned (e.g., shape, size, functional surface chemistry, circulating half-life, structural stability, permeability, and extravasation) [56]. Additionally, many receptor-facilitated contacts are required to maintain a high degree of surface conjugation chemistry and the related targeting capabilities. One of the primary mechanisms for BBB penetration is receptor-mediated transcytosis (RMT) [57], Transcytosis mediated by adsorption [58] and cell-mediated transport by immune cells, macrophages, and monocytes [59,60] are further examples. Extracellular vesicles, liposomes, and red blood cell membranes are just a few examples of the various nanoscale materials that have been used as nanomedicines thus far [61,62], in addition to metal nanostructures [3]. Drug-carrying nanostructures significantly enhance the pharmacokinetics and biodistribution of pharmaceuticals in the CNS when compared to free drugs [18,63].

Neurodegenerative diseases adversely affect the structure and function of neurons, which are a building block of the nervous system, and are frequently involved in neuronal death [64]. Mechanistically, the factors underlying the pathogenesis and propagation of neurodegenerative disorders include genetic mutations [65], mitochondrial dysfunction [66], damage to DNA or organelle membranes, protein misfolding and aggregation, autophagic as well as apoptotic cell death, and transglutaminase binding [67]. With a million people affected by Alzheimer’s disease [68], it is irreparable, though the available treatments consist of pharmaceutical, behavioral, social, and caregiving treatments. Pharmaceutical treatments include donepezil, rivastigmine, galantamine, tacrine (acetylcholinesterase (Ache) inhibitors), and memantine (N-methyl-aspartate (NMDA) receptor antagonist) [69]. The treatment of Parkinson’s disease (PD) is similarly critical, with 10 million people suffering worldwide. Treatments such as medicines, surgery, and physical therapy have been confirmed to increase quality of life. The primary treatment for extenuating motor symptoms is levodopa, used in combination with a dopa-decarboxylase inhibitor and other inhibitors such as dopamine agonists, catechol-o-methyltransferase (COMT) inhibitors, and monoamine oxidase-B (MOA-B) inhibitors [70]. Nanomedicine is unarguably moving with the anticipated pace to conquer neurodegenerative disorders and their associated consequences. Incidentally, nanomaterials have been manipulated to attain appropriate physicochemical properties, for instance the type of matrix, size, charge, surface chemistry, polarity, etc., to enable the crossing of the BBB and targeting the CNS to release their cargo [71]. Current intriguing advances in various kinds of polymeric nanoparticles, metal nanoparticles, nanoparticles (NPs), namely liposomes, etc., for the cure and identification of numerous neurodegenerative diseases are reviewed here.

This review presents collected information about different therapies, such as gene therapy, chemotherapy, and especially immunotherapy, based on nanomedicine for the treatment of CNS diseases. The diseases under study include brain cancer and the neurodegenerative Alzheimer’s and Parkinson’s diseases. The data are hypothesized from the experimental models of the oriented propagation hypothesis for glioma, tauopathies, amyloid-β, and α-synucleinopathy Lewy bodies. This study evaluates a role in disease progression, the latest information about different therapies, especially immunology, that have a role in disease progression, outlining the current advances and shedding light on some problems. Finally, the information may surprise researchers as to whether these selective targeting strategies and the elimination of symptoms engage the immunotherapy approaches among various pathological hypotheses associated with CNS diseases.

## 2. The Immune System in the Brain

Since the CNS lacks a lymphatic system and early destructive T-cell responses are driven by the parenchyma, it is usually viewed as an immune-privileged system [35]. According to the results of a recent study, despite the presence of a functional meningeal system, particles were able to move into subterranean cervical lymphatic nodes due to the functioning meninges [72]. Lymphatic endothelial cells were intensely studied for their configuration that expresses all their molecular markers [73,74], and, nowadays, they are generally recognized as the glymphatic system. The particular range of the system (Figure 1) is still unknown [75]. According to another study, the cerebrospinal fluid and interstitial fluid periarterial space containing macromolecules and solutes are continuously exchanged within the system of perivascular tunnels, basement membrane, and astroglia cells [76,77]. Additional research studies revealed a specific perivascular section for small lipid transport and glial communication signaling [78].

CNS immune treatment was partially credited to the absence of a standard lymphatic system. The parenchyma of the brain was able to maintain and potentiate the immune response induced by allografts that were recognized by an immune response in peripheral organs [79]. As previously mentioned, the brain parenchyma and meningeal part of the brain shows a wide variety of properties when compared to the BBB, starting from the fact that the meningeal blood barrier is more permissive than the BBB, so immune cells are allowed to flow separately through the meningeal spaces even under physiological conditions [73]. Even though studies suggested that a model in which a selective barrier, for example, the BBB, does not characterize the individual static structure capable of separating immune cells outside the organs, nonetheless they are slightly permissive entrances that control the passage of cells in particular circumstances, such as in some specific phenotypes. In this circumstance, the limit of the CNS is not the authority for elimination. As a result, the capacity to communicate with the active immune system is limited [80].

Through an internal recirculation mechanism connecting the cerebrospinal fluid with intestinal fluid, antigens pass from the brain to deep cervical lymph nodes through meningeal lymphatic vessels [75]. Nevertheless, the response is low; therefore, it is required for a considerable number of antigens or secondary signaling to generate this response. Alternatively, the cervical lymph nodes may possess the ability to moderate immune responses to CNS antigens and tolerance or reactivity in response to these antigens [73].

## 3. Challenges in Nanomedicine-Based Immunotherapy in the Brain

Human brains are among the best-protected organs in the body. The skull, meninges, and cerebrospinal fluid provide shields of protection: the BBB and BBTB [81]. All these layers protect the brain from injuries and prevent diseases. However, the protective layers lessen the entrance of therapeutic agents to the brain in a diseased state.

Despite research on many drugs or medicines, only a few drugs, such as rivastigmine, memantine, galantamine, tacrine, and donepezil [82], are currently used to support the clinical treatment of neurodegenerative disorders [1]. The CNS is the most complex and sensitive structure in the human body, securely packed and sealed by the BBB and BCFB [83,84]. Numerous transporter systems are the critical factors for regulating the CNS’s internal environment; they are located at the BBB and are capable of shipping elements through the BBB. For instance, the multidrug resistance protein named P-glycoprotein (Pgp) is a vital transporter belonging to the ATP-binding cassette transporter family [85].

Due to its multifaceted nature, there are numerous limitations in delivering targeted medicine from the blood to the CNS. There is a lack of data about the role of medicines, the half-life of medicines, or bio-presences for brain cells, their effects on the CNS, or the irregular and uncertain relationships between off-target medicines and receptors and enzymes. The complicated pharmacology of some medicines, neurodegenerative symptoms, the lengthy latency period, and incompetence of medicines, the subsequent development of infections, an incorrect number of medicines, the patient’s genotype, and diverse reactions to medicines constitute the instability index of assessed medications. In addition, with concerns related to the brain, the difficulties become even more difficult [86,87,88]. Here, the three main obstacles inhibiting drug transport into the brain are discussed.

### 3.1. The BBB

The CNS is a good site in regards to immunity due to being protected by the BBB, which plays a critical defensive role in the transportation of molecules, adjuvants, and immune modulators from the bloodstream to the brain [89]. Nevertheless, the CNS has immature T cells in blurred form, while T lymphocytes in active form circulate and can do so during their movement [38]. Moreover, animals that bear tumors and glioma patients’ normal infiltration process of cytotoxic T cells should be considered [90]. During neuroinflammation of the CNS, the brain uses microglia and pericytes as antigen-presenting cells (APCs), which often triggers neurological disorders [40]. Despite BBB disturbance, some non-enhancing regions of the infiltrating tumor where the BBB is unbroken limit therapeutic drug contact [41].

The presence of the BBB and BBTB protective shells around brain tumors is a significant obstacle to delivering therapeutic agents (Figure 2). The BBB, acting as a barrier, forms from a tight restriction of cerebral capillary endothelium, astrocytes, pericytes, basal membranes with infrequent transcytosis, and endocytosis [52]. Due to the overexpression of existing P glycoproteins on cerebral endothelial cells, utmost active efflux process penetration occurs [41]. Moreover, macromolecules of all kinds and large parts of smaller molecules, including anticancer drugs, cannot pass through the tiny and tight holes present in the connective cells of the CNS [41]. Thus, modern chemotherapies fail to reach their target owing to the BBB.

### 3.2. Blood–Cerebrospinal Fluid Barrier

After overcoming the BBB, the next obstacle is the blood–cerebrospinal fluid barrier (BCFB) that is assembled by the epithelial cells of the choroid plexus (CP), which systemically deals with drug molecules [88]. The choroid plexus epithelium (CPE) is a secretary epithelium that is one of the most effective tissue types based on secretory mechanisms involved in balancing cellular transport mechanisms [92]. Similar to the BBB, this epithelial barrier system separates the blood and CSF. The CPE is tightly preserved throughout form-adjacent cells found next to the CSF-facing surface. The CPE comprises multi-specific efflux transport proteins and detoxifying enzymes, both preventing entry of potentially lethal compounds into the CNS [93]. Several plans for drug delivery have been developed, most of which target the microvascular endothelium and are suitable for the BBB. In addition to the BBB, targeting the BCFB shaped by the choroid epithelium is important for the curing of CNS diseases [93].

### 3.3. Multidrug Resistance Proteins

In recent times, the BBB was considered a static lipid membrane barrier. Physical feature selectivity to a wide variety of circulating compounds by the BBB membrane was due to factors such as cerebral endothelial cells containing tight junctions, deficiency of vesicles or caveolae, and high electrical resistance. However, immunocytochemistry, molecular biology, and biochemistry experiments on BBB transport pathways, have established that cerebral endothelial cells acquire a wide range of metabolic enzymes (e.g., glutathione transferases, alkaline phosphatase, and cytochrome P450 enzyme) and efflux transport proteins (energy-dependent), including P-glycoprotein and multidrug resistance proteins arranged irregularly and serving as a barrier. P-glycoprotein (P-gp) (170–180 kDa) is a plasma membrane-associated protein and is one of the essential ATP-dependent efflux transporters in the BBB. P-glycoprotein is a multidrug resistance protein encoded by the MDR1 gene [94], that limits the access of various drugs and preventing them from achieving their precise therapeutic objectives. In comparison, when P-gp expression is suppressed, the drug quantity arriving in the CNS was increased. Another experiment suggested that inhibitors and inducers of P-gp have an apparent effect on brucine pharmacokinetics and pharmacodynamics [94].

The challenge in pharmaceutical research is discovering devices and methods for allowing efficient and effective drug distribution into the CNS [95]. Immunotherapy is an approach of clinical oncology with significant development for effective therapies against numerous cancers [96]. However, the microscopic immunological environment in the CNS requires appropriate precautions to engage in immunotherapeutic strategies for gliomas. Immunotherapy has the potential for the treatment of glioblastoma (GBM), although researchers agree that combination regimens are required to attain a robust response. Owing to its powerful adaptive capabilities, relative lack of immunogenicity, immunosuppressive tumor microenvironment, and intra-tumoral heterogeneity, the amelioration of GBM across the BBB is still challenging. It is believed that the CNS has a unique, activated, distinctive, and adaptive immune system, with microglia cells specified for antigen presentation, expression of Toll-like receptors (TLRs), and complement components [97,98]. The CNS is also considered an “immunologically inactive” site with an environment employed to express immunosuppressive mediators [42]. Based on these facts, CNS disorders are a significant cause of delayed or hampered immune responses.

The engagement of multiple mechanisms to induce immunosuppression provides an obstacle limiting the value of immune therapies for glioma. The gloom remains as immunologically cold tumors with a suppressed TME are insufficiently infiltrated by functional immune effector cells. The key factors that modulate the immunologically cold microenvironment for T cell regulation, also called “Tregs”, in gliomas, are due to highly expressed markers such as CD^25+^, CD^4+^, and FOXP^3+^. Moreover, tumor cell-derived cytokines act as immune inhibitors, for example IL-6, IL-10, and transforming growth factor-beta (TGF-β) [44]. Glioma immune responses are due to the Tregs in both tumor-resistant or glioma-grade increased programmed cell death protein-1 (PD-1) expression [43,46]. Although there are existing tumor-infiltrating lymphocytes in the TME, it still has a very low response rates because of immunosuppressive signals from accumulation of exhaustion [45].

## 4. Nanomedicines for Targeting CNS Diseases

Neurodegenerative disorders such as Alzheimer’s (AD), Parkinson’s (PD), and amyotrophic lateral sclerosis (ALS) are characterized by a protracted pathological degradation process that causes incalculable suffering to patients and their relatives. With the maturation of nanotechnology since the 1990s, the technical barriers to nanomaterial research have disappeared. Nanomaterials have steadily entered the area of view of neurodegenerative disease experts. For research purposes, numerous nanomaterials, including lipid-based nanomaterials (e.g., liposomes and solid lipid nanoparticle (SLN)), polymeric nanomaterials (e.g., micelles, dendrimers, nanocapsules, and nanospheres), and inorganic nanoparticles, have been utilized. Nanoparticle-based therapeutic strategies in CNS-associated diseases are mainly focused on and sustain local release and targeted delivery of curative agents at the brain’s affected zone after bypassing the BBB [99]. The hallmark feature of AD is amyloid-β peptides combining to form amyloid plaques. Existing AD treatments include cholinesterase inhibitors like donepezil, rivastigmine, galantamine, and N-methyl-d-aspartate receptor antagonists (memantine) [100]. With the help of polymeric nanoparticles, lipid-based nanoparticles, and non-polymeric quantum dots, passage through the BBB is permitted and side effects of free drug usage are decreased by reformulating clinically used drugs [101]. Concerning the nano delivery method, numerous neuroprotective compounds such as metal chelators and several NMDA antagonists of anti-amyloids improved the passage through the BBB and reduced Aβ aggregate formation [102,103].

PD’s treatment based on dopamine replacement is the most used approach; later on, treatments for the effects on motor neurons and attenuating disease development were developed, although these drugs’ effects on behavior and perception are still under investigation [104]. Recent studies focused on curative nanoparticles in various approaches, for example, targeting dopamine delivery by using polymeric nanoparticles or liposomes [105]. Many studies use different drugs such as bromocriptine, apomorphine, mitoapoacynin, and ropinirole encapsulated with liposomes or polymeric nanoparticles to advance the sustained discharge of drugs and to decrease unwanted outcomes of predictable PD therapy [106,107,108]. Anti-inflammatory therapeutics have been established via PEGylated liposomes or polymeric nanoparticles to inhibit neuronal cell death in PD [82,83,84] and increased dopamine levels [109,110]. Alternatively, numerous groups proposed nano-systems for the transfer of hereditary material such as nucleic acid (DNA and RNA) and oligo-nucleotides to prevent abnormal gene expression or production of beneficial proteins in targeted cells [111]. Significant progress in clinical signs was noted in progressive PD patients using gene therapy, which is still a contradictive matter considering PD’s heterogenic pathology [112].

Recently, nanomedicines based on GBM therapeutics have received important considerations [101,113]. In the case of generally directed free drugs, a minute proportion passes through the BBB in a nonspecific manner in off-target tissues, resulting in severe undesirable outcomes. Therefore, nanoparticle use for drug delivery to the brain can increase the percentage of drugs crossing the BBB and decrease nonspecific drug accumulation in other tissues [15]. For instance, gadolinium-loaded nanocarrier systems improved drug penetration and targeting ability to 100-fold higher levels than free gadolinium. NPs with a ligand-modified surface have facilitated imaging of brain tumors [114,115]. In another study, the PEGylation of nanoparticles was used in drug delivery to protect nanoparticles from blood protein interactions on the reticuloendothelial system (RES) [102,103]. Similarly, the PEGylated NPs thoroughly avoided interfacing with protein in the plasma. Studies have found, however, that PEGylating proteins does not completely prevent their interaction with nanoparticles in the blood [116].

Nanomedicine-based approaches have the promising potential to overcome these restrictions and advance the treatment of CNS disorders. The uses of nanoparticles as drugs and drug-delivery carriers have been extensively examined in preclinical studies and are currently being applied in the clinical setting for certain CNS diseases (Table 1) [117,118,119,120,121]. A variety of nanoformulations have demonstrated that they significantly improve CNS pharmacokinetics and distribution in brain areas when compared to free drugs [122,123]. In addition to overcoming the BBB, surface decoration with specific ligands allows ‘active targeting’ to different brain cell types [101]. Targeting neuronal cells in CNS pathology markers is a crucial task in nanomedicine and neuroscience. Presently, there is only a limited number of nanomedicines in clinical use or in the pipeline to cure CNS disorders, such as glatiramer acetate [124], peginterferon-1a for sclerosis [125], gold nanocrystals (CNM-Au8) for lateral sclerosis and PD [126], APH-1105 for AD [119,127], and nanocurcumin for lateral sclerosis [128,129]. These nanomedicines are mainly based on polymeric-based therapeutics, gold nanocrystals, and curcumin encapsulated in nanomedicine in nano micelles, or NPs containing an inhibitor of α-secretase to treat Alzheimer’s disease. Although these nano-formulations were not precisely made to target neurons, their systematic features and the changes required to enhance their clinical results, are still poorly known. The low number of nanomedicines used to target the CNS in clinical trials is the main difficulty facing the strategy and advancement of these methodologies. Thus, a comprehensive preclinical confirmation is mandatory. Therefore, in recent years, numerous struggles have concentrated on the advancement and testing of nanomedicines in models of CNS disorders that can advance the clinical usage of these NPs in the CNS.

## 5. Surface Modification of Nanoparticles

The nanomedicine system used for CNS diseases must be compatible and able to cross the BBB efficiently. While designing nanocarriers, the following parameters require optimization: size, shape, structural stability, functional surface chemistry, permeability, circulating half-life of the tumor, extravasations, and targeting capability to enter the tumor using receptor- or transporter-facilitated interactions. In this aspect, surface-conjugated chemistry may play a pivotal role in overcoming the significant challenges traversing the BBB. The BBB is a group of highly specialized, closely connected cells arranged in a group. The BBB can protect the brain from many potential threats and it has a pivotal role in the brain’s homeostasis and physiology. The BBB’s structural features primarily consist of highly impenetrable brain capillary endothelial cells (BCECs) enclosed with the wrapping of perivascular end feet of astrocytes and basal lamina pericytes (as shown in Figure 1A) [56]. The epithelial cells possess tight junctions (TJs) that strictly restrict the permeability of water-insoluble molecules via diffusion. Thus, the BBB acts as a barrier resisting nearly all (98%) small-sized adjuvants and drugs, whereas the large-molecule neuro-therapeutics are completely blocked [156].

Luminal and abluminal epithelial cells act as transport sites for substrates into the brain [56,156]. Lipophilic molecules (≤400 Da), alcohol, nicotine, steroid hormones, CO_2_, and O_2_, can take a transcellular lipophilic pathway to cross the BBB via diffusion. Detailed routes are shown in the schematic in Figure 1B. Moreover, carrier-mediated transcytosis (CMT) is the primary method to cross the BBB that is used by the many essential nutrients and ions, such as amino acids, glucose, nucleosides, vitamins, and electrolytes. Glucose transporter-1 (GLUT1) is another essential receptor that favors bi-directional glucose diffusion via a concentration gradient [57]. Furthermore, another pathway includes receptor-mediated transcytosis (RMT), which includes some specific receptors, such as the receptors for low-density lipoprotein [157], insulin, and transferrin, as natural targets for nanomedicines [158]. Meanwhile, adsorptive-mediated transcytosis (AMT) used to transport molecules including cationic proteins or cell-penetrating peptides (CPPs) [58], can interact with anionic sites on the surface of the epithelial cell membrane through electrostatic interactions to transport the payload. Moreover, transport across the intact BBB can be achieved in a cell-mediated manner, including stem cells and immune cells, such as monocytes and macrophages [59,60]. In some cases, the payload gets into the CNS. The nanomedicine-based delivery system may enter the endothelium by efflux pumps but the drugs may be squeezed back into the circulatory system. This type of system is one of the protective mechanisms that occur naturally in the brain and are responsible for avoiding exposure to foreign molecules or materials. There are two major transport carriers, including multidrug resistance proteins (MRPs), such as adenosine triphosphate (ATP)-binding cassette and P-glycoprotein (P-gp) [53], and convection-enhanced diffusion (CED), which have been the most extensively exploited for clinical studies based on convection for drug delivery with the aid of constant hydrostatic pressure gradients [54].

## 6. Applications of Nanotechnology in CNS Disorders

The development of treatments for neurological disorders also includes experts in nanotechnology. The field gives new techniques to treat PD, HD, AD, stroke, brain tumors, and epilepsy. Molecules are designed in such a way that they are able to cross the BBB; specific cells are targeted or used as a pathway for the signaling process. The designed molecules are used as carriers to deliver genes. In addition, experts in nanotechnology have started work on delivering radio-contrast objects to aid in diagnosing diseases through imaging.

### 6.1. Glioblastoma

Glioblastoma and gliomas are inborn brain tumors that arise from the ancestor cells of neuroglial [21]. Due to their localization, morphologic resemblances with non-malignant neuroglial cells, and propagation, immunotherapy is extremely challenging for individual brain cells [159]. Due to these facts, this type of disease might be included in a diverse range of CNS cancers, including astrocytomas, oligodendrogliomas, ependymomas, neuroblastomas, and glioblastomas [159,160,161]. GBM is the most common CNS tumor in adults, comprising 50% of all gliomas and 15% of primary brain tumors [23]. The overall median survival rate is less than two years, even with antagonistic therapy [21]. The U.S. Food and Drug Administration (FDA) has approved a wide range of oncology drugs for treating different types of cancers in recent decades; nevertheless, only two drugs have successfully passed the clinical trials for glioma therapy, temozolomide and the anti-angiogenic drug, bevacizumab [22].

Many researchers have tried to develop new nanotherapeutic approaches for treating gliomas. Nano-formulations were developed that deliver drugs across the BBB [162]. One nano-formulation that involves PBCA mixed with methotrexate [163], and another using temozolomide [152], produced significantly improved results for delivering drugs into the brain compared free drugs. Similarly, in vitro results using SLNs containing paclitaxel [153] showed an increase in inhibitory effects on the proliferation of glioma cells [164]. It was also demonstrated that transferrin receptor-targeted nanoparticles can enhance cellular internalization and cytotoxicity of docetaxel with enhanced pharmacokinetics [165,166]. Dendrimers are used to transfer antineoplastic treatments into the brain. A conjugated molecule made of polyether-copolyester (PEPE) dendrimers and methotrexate confirmed improvements in cytotoxicity in cultured U343 and U87 cancer cell lines. The nano-formulation was able to overcome resistance from drugs [167]. The conformation from PEGylated PAMAM dendrimers with doxorubicin (DOX) opens a new therapeutic window by hindering C6 glioma spheroid proliferation. There was little cytotoxicity in opposition to brain micro-vascular endothelial cells in vitro [168].

A contrast agent NP that offers detailed cellular and molecular imaging can aid in the surgical removal of gliomas. In a study conducted by Hernandez-Pedro et al., gadolinium oxide crystals of a size less than 5 nm in diameter were used to label glioma cells [169]. In another study, researchers used an antineoplastic drug encapsulated in PEG-coated hexadecyl cyanoacrylate NPs in a mouse model of gliosarcoma to study the precise drug-release kinetics and found higher diffusion of the drug through the BBB, with regard to bulk drugs [170].

Over the last several decades of research on understanding malignant brain tumors’ etiology, there has only been limited successes in the development of treatments. Typically, glioma patients are treated with surgical re-section of the tumor site, chemotherapy, and radiation, which can result in severe side effects [171]. Due to the CNS’ structure, the tumor’s location reduces the efficiency and preciseness of therapeutic interventions to cure the disease entirely; therefore, recurrence is inevitable. In 2005, temozolomide was widely implemented for chemotherapy of newly diagnosed glioblastoma, yet uncertainty has been found in the survival of glioblastoma patients [172,173]. Newly diagnosed GBM with full resection of the tumor is treated with temozolomide and radiotherapy [23]. In most patients, GBM resulted in a worsening effect and did not ensure a healthy life after relapse [174]. Therefore, the light treatment of patients with GBM are considered and significant interest is being directed to develop new therapy techniques for this disease.

#### 6.1.1. Cellular Immunology for Malignant Gliomas

To date, even with decades of extensive experimental work, only a modest increase in patient life expectancy has resulted, even in the most promising clinical trials. The mechanisms underlying these small developments are still poorly understood. Interestingly, GBM possesses unique immunosuppression mechanisms; for instance, GBM infrequently shows metastasis to extracranial sites, even though circulating tumor cells (CTCs) have been detected in GBM patients [175,176,177]. Glioma grows surrounded by the BBB and BBTB that provides a reliable shielding mechanism from the immune system. Hence, it is also challenging for the transmigration of immune cells [178]. However, when T cells are in an inactivated state, they can cross an intact BBB and BBTB [179]. Furthermore, the BBB is compromised in GBM because of amplified fenestrations, disruptions in tight junctions, and low BBB-connected pericytes [180,181]. Therefore, the success rate of immunotherapy of glioma was limited until now. Gliomas with reduced anti-glioma immune responses expressed interleukin-10 (IL-10) and TGF-β [182]. For instance, TGF-β constrains the proliferation pathways and T cell activation, suppresses lytic enzymes production, and initiates the expansion of naive T cells into regulatory T cells [183,184]. Under physiological conditions, regulatory T cells are needed to shield against autoimmune diseases. Nonetheless, Tregs have been suggested to be the foremost contributor to depressed cellular immunity in glioma patients [185]. Furthermore, vascular endothelial growth factor (VEGF) and basic fibroblast growth factor (bFGF) are considered indispensable mediators in glioma angiogenesis and can induce expression of the inhibitors for cell adhesion molecules [22] in non-glioma cells in both animal models and endothelial cells extracted from normal tissues [186,187,188,189]. Dendritic cells (DCs), which originate from myeloid cells, are the most potent APCs and usually represent tumor-derived epitope peptides. Furthermore, these cells play an essential role in producing histocompatibility-complex (MHC) peptides. When MCH peptides can activate T cells, which then expand clonally and travel to the tumor-containing organs where they recognize antigen epitopes on tumor cells in an MHC/peptide complex similar to the TCR [98]. Thus, tumor cells are guided to activated T cells, which can release preformed cytotoxic molecules such as granzyme and perforin [190]. A more detailed schematic illustration is shown in Figure 3.

#### 6.1.2. Vaccine-Based Immunotherapy of Glioblastomas

Numerous vaccine-based immunotherapy techniques, such as altering glioma cells, dendritic cell application, peptide-based vaccinations, and combination approaches with various therapeutic modalities, have been explored extensively in rodent models [192]. The most prevalent targets in glioblastomas are mutant IDH, EGFRvIII, a panel of antigens, or personally selected antigens. In forty percent of glioblastoma cases, the EGFR gene is amplified, and in more than fifty percent of cases, exon 2–7 is deleted or mutated [192,193]. The mutated form of the protein lacks a ligand-binding domain, resulting in constitutive activation, which in turn promotes cancer. Likewise, the mutant receptor can be activated by a variety of kinases, including Src family kinase [193]. The altered amino acid sequence has been revealed to be immunogenic. A vaccine called rindopepimut, generated using the mutated peptide sequence, was developed to prompt the immune system [192]. The early clinical trials indicated benefits from the vaccine. However, the subsequent phase III clinical trials failed to indicate benefits in a general population [157]. In another clinical trial, named ReACT, patients with recurring glioblastoma received a control or rindopepimut. The vaccine presented benefits over the control, as the medium survival rate was 12 months versus 8.8 months, respectively [194]. Utilizing NPs can increase the efficiency of the vaccine. They can protect the vaccine against degradation and boost the absorption by APCs. Kuai et al. created nanodiscs using lipids and peptides derived from high-density lipoprotein. Antigen peptides and cholesterol were attached to the surface at that time. In mice with melanoma tumors, the nanodiscs were more effective in priming T cells and the tumors took longer to manifest. In tumor-bearing mouse models, nanodiscs plus anti-PD-1 treatment inhibited the progression of cancer in 88% of animals, which is much greater than for either treatment alone [195]. Vaccines can also be administered as mRNA. Liu et al. created lipid/calcium/phosphate nanoparticles with MUC1 mRNA. The immunizations were subsequently administered alongside the anti-CLTA-4 antibody to mice with triple-negative breast cancer. The dual treatment provided a superior response than either treatment separately [196].

#### 6.1.3. Cargo-Loaded NP-Based Immunotherapy of Glioma

The delivery of therapeutic agents based on cargoes that induce immune responses for glioma treatment is fascinating. An extended narration highlighting this topic is given in the recent online article “Glioblastoma is ‘hot’ for personalized vaccines” [197]. In these aspects, nanotechnology may perhaps play a vital role in engaging immune cells. The current barrier in glioma therapy can be tackled with nanomedicines based on immune-therapy using nanoparticles [198]. This study has critical shortcomings, lacking the induction of cytotoxic anti-tumor responses from T cells, rather than merely drug delivery in tumor diseases. Therefore, a practical, precise cancer vaccine could be developed via administrating immune-modulatory antigens and agents. Moreover, the cancer vaccines capable of targeting and activating T cells directly are biological compounds and molecular cargoes such as immune checkpoint inhibitors, suppressors of TME modulation, RNAi, nucleic acids, and adjuvants [199,200,201]. So far, different nanostructure materials have been used as cargo-loaded adjuvants and drugs (presented in Figure 3). Herein, various targeting steps of nanomedicines are described using different strategies for activating the immune process in the cell-mediated immunity targeting glioblastomas and other CNS tumors.

Babak et al. [202] developed nano-tubes with multiple layers of carbon. The tube delivers siRNA and DNA cargoes into GL261 glioma and BV2 microglia. With an inert nature, the nanoplatform is also biodegradable and non-toxic and has advantages for brain tumor immunogenetic therapy. Moreover, APCs can efficiently internalized nanoparticles to develop an immune-stimulatory cascade in the brain’s cancerous cells [203].

Recently, many therapeutic approaches have been designed to synergistically treat gliomas. For instance, Qiao et al. [204] have replicated the nano-theragnostic response of reactive oxygen species (ROS) polymers of nanoparticles. The compounds loaded with Angio pep LipoPCB (TMZ+BAP/siTGF-β) (ALBTA), which is composed of iron oxide NPs (IONPs), TMZ drug, and siRNA targeting TGF-β, an immunosuppressive cytokine. For crossing the BBB, Angiopep-2 was changed on the surface of the NPs. The overall endurance of mice was increased by improving the immunosuppressive micro-environment. The frequent correlation of expression of vascular laminin-411 (α4β1γ1) with a tumor with a higher expression of cancer stem cell markers (e.g., Notch, CD133, nestin, c-Myc) shortens the lifespan of GBM patients. A nano-bioconjugate that inhibited laminin-411 crossing the BBB, inhibited markers of stem cells, and targeted the TME resulted in the increased survival of mice with cephalic cancer [205]. On the other hand, in immunotherapy, the low accruement of glioma antigens by APCs acts as a barrier. Recently, to address this issue, a “cluster bomb” nano-vaccine was developed with a high-loading-capacity antigen carrier with zinc oxide and triblock-copolymer nanoparticles on the surface [206]. The main chain reduction-sensitive polymer MPSDP can react with dithiopyridine in the Polydopamine (PDA) that is blocked by the sulfhydryl group; the self-assembled MPSDP polymers resemble “cluster bomb” nano vaccines, which are then spoiled by the hydrophilic–hydrophobic interaction. Due to the existence of multiple interactions in the nano-vaccines, the adjuvant to cellular and humoral immunity is promoted. The nano-vaccine with a three-fold reduction response can trigger the vaccine to bomb in antigen-presenting cells. Cytotoxic T lymphocytes and antibody responses to cytokine secretion is promoted by CD^8+^ vigorously. A multi-dimensional platform used for nanoparticles improves the accumulation of drugs targeting the immune system at the cellular and humoral levels, which improved the survival of mice with tumors [206]. The combination of T-cell activators that have anti-tumor activity, such as α-galactosyl ceramide, with C6 glioma-derived exosomes to treat rat models with GBM was investigated by Liu et al. [207]. The induction of an amenable immune response, including expression of IFN-γ and TNF-α, was increased after vaccine-based immunotherapy.

The nanotechnology approach is combined with therapy using chimeric antigen receptor (CAR)-T cells. This approach is the cutting edge of treatments of solid tumors. For instance, Zhang et al. [208] proposed that the TME assists iRGD-lipid nanoparticles in upgrading the perseverance and function of transferred CAR-T cells in glioma patients. An objective of lipid nanocarriers is to induce therapeutics in glioma tumors. On one hand, it can remove the protumor cell populations (“releasing the brakes”), while on another, it can stimulate the essential antitumor effector cells (“stepping on the gas”). Recently, Shi et al. [209] developed polymersomes coated with Angiopep-2 and anti-PLK1 siRNA payloads. In mouse studies, it has not only improved BBB permeability, but also greatly stimulated anti-GBM activity. In addition, another study demonstrated that folate-targeted polymeric micelles can deliver TMZ and anti-BCL2 siRNA in rat models with orthotopic glioma. Intracerebral administration of this nanocarrier-based combination therapy inhibited tumor development and prolonged survival [210].

Recently, Galstyan et al. [211] demonstrated the utilization of a poly (-L-malic acid) natural biopolymer scaffold to which a-CTLA-4 or a-PD-1 nanoscale immunoconjugate targeting moiety was covalently bonded (NICs). NICs are utilized to systemically transfer payloads across the BBB and to stimulate local anti-tumor immune responses in the brain. NICs were used to treat intracranial GL261 glioblastoma (GBM) with an increase in CD8+ T cells, NK cells, and macrophages, and a decrease in regulatory T cells (Tregs) in the microenvironment of the brain tumor. On the one hand, the tumor-targeted polymer-conjugated NICs function as checkpoint inhibitors as a prospective GBM treatment by stimulating the systemic and local brain tumor immune responses, which are associated with increased survival in mouse models. Therefore, the proliferation of tumor cells is attenuated, resulting in enhanced survival rates.

#### 6.1.4. Nanomedicine-Based Combination Therapy

The complex immunosuppressive tumor microenvironment (TME) in glioblastoma overwhelms endogenous antitumor immune activity, resulting in increased immunological tolerance [212]. Recently, immune-stimulating non-methylated oligonucleotides (such as CpG) have been created to tackle this issue, boosting long-term immunity against resistant tumors [213]. By preventing long-term relapses, Lollo et al. have achieved the highest therapeutic index against GBM by combining chemotherapy with immune-stimulating CpG-mediated immunotherapy [214]. PTX/CpG co-loaded lipid nanocapsules dramatically increased the survival rate of orthotopic GL261 glioma-bearing mice compared to single-loaded PXT in the same system without CpG. Consequently, they noticed a greater anti-glioma efficacy with the combined chemotherapy and immunotherapy than with chemotherapy alone.

Recent studies have also shown that specific tumor cells escape the immune system’s elimination through modulation of immune checkpoint pathways [215]. indoleamine 2,3-dioxygenase (IDO) is abundantly expressed in brain tumors and is recognized as one of the most important immune checkpoint receptors. It is a key immunotherapeutic target for numerous brain cancers [216,217]. Recently, 1-methyltryptophan (1MT), which is a specific competitive inhibitor of IDO, has been shown to inhibit IDO expression and slow down tumor cell growth [218]. However, to date, 1MT has not been proven as an efficient targeting agent. Therefore, few research groups have combined 1MT with other drugs for effective targeting [219]. For instance, Kuang et al. [220] developed a nano-strategy of co-delivery of 1MT with DOX on mesoporous silica NPs (MSNPs) modified with tumor-targeting/penetrating peptide CRGDK/RGPD/EC (iRGD) for therapeutic application to orthotopic gliomas.

Recently, Kadiyala et al. [221] created nanodiscs modified with a high-density lipoprotein-mimicking nano-formulation (called sHDL nanodiscs) that could not only carry glioma tumor-specific antigens, but also operate as a vehicle for the administration of adjuvants and bioactive compounds. To target glioma both in vitro and in vivo, sHDL nanodiscs were loaded with CpG deoxynucleotides, a Toll-like receptor 9 (TLR9) agonist, and a chemotherapeutic, namely docetaxel (DTX). The combined chemo-immunotherapy delivery system utilizing DTXsHDL–CpG nanodiscs not only targeted the drugs to the tumor site, but also activated antitumor immune responses and prevented the recurrence of the tumor due to the increased delivery of bioactive compounds to immune cells surrounding the glioma. The delivery of payloads in DTXsHDL–CpG nandiscs into the tumor mass induced tumor relapse and improved antitumor CD8+ T cell responses in the immunosuppressive TME of the brain. The standard of care for glioblastoma multiforme (GBM) combined with the DTX-sHDL-CpG treatment led to tumor degradation and considerable improvement in the median survival of glioma-bearing mice up to 80%. In addition, galectin-1-targeting siRNA encapsulated in an intranasal chitosan nanoparticle is a promising option for enhancing the efficacy and dependability of chemotherapy and immune checkpoint suppression in order to increase the life expectancy of mice with tumors [222]. This intranasal siGal-1 delivery induced a dramatic alteration in the composition of the tumor microenvironment (TME), including a decrease in myeloid suppressor cells and regulatory T cells, and an increase in CD4+ and CD8+ T cells, during the progression of GBM. A therapy combining siGal-1 with TMZ or an immunotherapeutic strategy (such as dendritic cell immunization and PD-1 blockade) exhibits synergistic effects.

#### 6.1.5. Gene Therapy for Glioma

Gene therapy appeared as a new treatment for many human cancers. It could be specially used to control the oncogenes in anti-tumor treatments [223,224]. The adenovirus-facilitated gene therapy with sitimagene, ceradenovec, and ganciclovir, and later resection, improved the survival time of patients with recently identified glioblastoma multiforme [225]. Therefore, gene therapy was commonly offered as a valuable support for recent glioblastoma treatments [226]. The combination of gene therapy with immunotherapy must overcome the full consequences of glioblastoma cure [227,228]. For example, Maria-Carmela Speranza et al. used a non-replicating adenovirus with the HSV TK gene to potentiate the anti-PD-1 effect in a syngeneic glioblastoma mouse model. The HSV TK, also known as AdV-tk, increased PD-L1 levels and cytotoxic CD8+ T cells were induced to localize in the tumors. This combination therapy increases the survival of animals from 30–50% to 88% [229]. Although a possible gene therapy, the basic features of this gene restricted the effective transfer to cancer locations and delayed their development for clinical uses [230,231].

Transport across the BBB presents an additional obstacle for anti-glioma therapies. Nanoparticles can compensate for gene deficits and carry gene and immunotherapeutic drugs to brain tumors safely. Gulsah et al. developed a cyclic peptide iRGD ornate solid–lipid nanoparticle to transport siRNA against EGFR and PD-L1 for combination therapy. The siRNA-targeting nanoparticles decreased EGFR levels by 54.7% and PD-L1 by 58.6%. Additionally, the average lifespan of mice treated with f(SLN)-iRGD: siRNA by radiation increased to 38 days. These three combinations significantly increased the survival rate of mice.

Gene therapy was frequently used to modulate immunosuppressive signals and enhance systemic therapy in glioma treatments, according to the available evidence. These could increase the immune response to immunotherapy in a synergistic manner. However, there are few published studies on this strategy. Current anti-glioma treatments typically include gene therapy and immunotherapy as supportive therapies. Moreover, the paucity of genes and antigens remains the most significant hurdles to an effective therapy.

#### 6.1.6. Chemotherapy

Historically, Egyptians were the first to use chemotherapy, and proper exploration of chemotherapy started when some soldiers in World War II who were exposed to some chemicals started showing unusual reactions. Currently, chemotherapy is one of the most established clinical therapies, which is due to its convenience and several chemotherapeutic drugs are FDA-approved [232]. Chemotherapeutic drugs disrupt cellular function at one or all phases of the cell cycle. Some of the key chemotherapy agent categories are: antimetabolites, plant alkaloids, alkylating agents, and anti-tumor antibiotics. As cancer is a complex disease, multiple complicated mechanisms are involved, such as the growth, progression, and invasion processes [233].

CNS tumors are neoplasms arising from numerous kinds of cells within the CNS, which account for 2% of all cancers. Every year, out of a hundred thousand, about nineteen individuals are diagnosed with primary brain tumors and CNS tumors, worldwide. For example, GBM is one of the utmost violent and common cancers occurring in the central nervous system. In 2005, a study reported a 27.2% survival rate with oral temozolomide (TMZ; an alkylating cytotoxic agent) and concurrent radiotherapy, while the survival rate was only 10.9% with radiotherapy alone [234].

However, accumulating data suggest that, in the case of single curative strategy, it results in drug resistance and gives rise to tumor cell tolerance, leading to tumor recurrence and metastasis. Considering the associated complications, combination chemotherapy is being used in several cancers. Therefore, therapeutic strategies combined with various agents should be able to overcome these problems. Still, these therapeutic agents have some shortcomings, such as insufficient accumulation, controlled transportation, and release at the tumor site and short half-life in circulation [235].

In GBM treatment, to overcome physiological barriers and potentiate the therapeutic effects, nanotechnology has been investigated as a new approach. For instance, delivery systems that have been investigated include: liposomes, lipopolymer nanoparticles, dendrimer nanoparticles, polymer nanoparticles, and hybrid nanoparticles [236]. In the case of GBM, nano-delivery system transport of multiple therapeutic agents across the BBB mediated by adsorptive- or receptor-mediated endocytosis or carrier-mediated transport, can improve tumor targeting in the brain and reduce side effects [237]. In addition, the introduction of stimuli-sensitive responses into delivery systems can ensure the maximal drug retention at the desired sites [238]. In the last few decades, intensive progress has revealed effective and promising results in cancer therapy, but only a small amount of them have been applied in clinical trials.

### 6.2. Alzheimer’s Disease (AD)

The most common neurodegenerative diseases are AD and PD, from which millions of people are suffering worldwide [239]. Regardless of all the scientific advancements, the currently available therapeutics have low efficiency in the amelioration of these diseases. The significant hurdles come with halting disease progression and late-stage diagnosis. In contrast, the only symptomatic treatment without modifying these diseases’ progress is the current state-of-the-art therapeutics. The causes of AD are known to be either extracellular amyloid-beta (Aβ) peptides deposition in senile plaques or the formation of neurofibrillary tangles with phosphorylated tau proteins [158,240]. Another pathological hallmark includes the tauopathies caused by hyperphosphorylation of tau protein deposits and its insoluble aggregates inside neurons in CNS disorders [241]. As a result, the AD-suffering patient may progressively lose memory, develop problems with proper functioning in a physical environment, fail to make decisions, and develop language difficulties as the most domineering clinical hallmarks of this disease [242].

Many nano-formulations have produced positive results in their impact on AD patients. The conformation using PEG stabilization of nano-micelles comprised of lipids with phosphate lessens the aggregation of Aβ, devalues neurotoxicity induced by Aβ in human SHSY-5Y cells, and an in vitro neuroblastoma cell line [243]. An experiment in mice showed meagre bioavailability, even though the phytochemical curcumin was able to lessen cytotoxicity and oligomerization [244]. The bioavailability enhanced, but did not harm, Aβ’s aggregation ability by non-liposomal deviation of curcumin [245]. Excess metal ion-like copper usage plays a role in the pathology of AD; thus, chelating agent usage is another technique for managing AD [246]. Microemulsion of nanoparticles with copper d-penicillamine as a chelating agent showed a profound ability to cross the BBB and soften Aβ aggregates in vitro [247]. Furthermore, in the pathology of AD, oxidative damage is another key factor that suggests the application of antioxidants in the management of AD. It has a neuroprotective effect against glutamate receptors that induce excitotoxicity. This can be achieved using derivatives of fullerene with the ability to act as scavengers of free radicals. In addition, neuroprotection by fullerene against Aβ toxicity has not yet been proved. However, an ability to inhibit Aβ peptide fibrillization and prevent Aβ from inducing cognitive injury after intraventricular administration suggests a beneficial role for fullerene for AD treatment [248].

Another feature of the pathology of AD is the obvious scarcity of the neurotransmitter, acetylcholine (ACh). ACh breaks down promptly in the blood if it is directly administration to the body. An approach using nanotechnology is to deliver ACh directly to the brain to balance the ACh levels. Nanotubes filled with ACh significantly restored cognitive function to pre-AD levels in a kainic acid mouse model compared to free Ach [249].

Generally, Aβ peptide is produced by several CNS cells, including neurons, as the result of the proteolysis of amyloid precursor protein (APP) [250]. The β- and γ-secretases are responsible for cleaving APP to form Aβ fragments with differing numbers of amino acids, ranging from 36 to 42 [251]. Still, the exact origin of the formation of Aβ peptides, which can accumulate in the brain, is not understood [252]. However, some transport molecules can be transported across the BBB with the help of the apolipoprotein family (Apo-E2/3 and Apo-E4) and two efflux pumps [253,254]. Moreover, the LRP and the very-low-density lipoprotein receptor (VLDLR) [255] are the primary transport methods to cross the BBB.

The available approved drugs for AD treatment include donepezil, galantamine, and rivastigmine, which can target metabolic shortfalls [256]. Nevertheless, these drugs are also associated with brain function impairment. Typically, these drugs may be classified as either targeting the acetyl-cholinesterase inhibitors (AChEI) or N-methylD-aspartate (NMDA) receptor [257]. The deposition of Aβ triggers a hypothetically distinct pathological immune response in AD [258]. Interestingly, the microglial cells have natural machinery for removing protein aggregates and debris from the brain [259]. Several routes play a role in the clearance of amyloid-β from the resident microglial cells. The activation of microglial cells might clear the amyloid-β after immunization, as after a stroke or amyloid-β injection [260]. The clearance of amyloid-β is associated with higher microglial cell activity [261]. Recently, reports suggest that, the mice that overexpressed transforming growth factor-β (TGFβ1) were crossed with *APP*-transgenic mice to produce offspring with reduced amyloid loads and amplified microglial cell activation [262]. Moreover, blocking of complement evasion or activation of pathways to decrease microglial cell activity has also been observed in *APP* transgenic mice [263]. A detailed description of how immune modulation may play an important role in degrading and clearing NFTs is well addressed by Winner et al. [264].

Recently, Liu et al. reported applying a zwitterionic poly(carboxybetaine) (PCB)-based nanoparticle co-loaded with fingolimod, siSTAT3, and zinc oxide into the polymeric NPs named as MCPZFS NPs [265]. The NPs inhibited microglia and Aβ recruitment for the effective treatment of AD. The MCPZFS NPs significantly improved microglia priming by reducing the level of proinflammatory mediators and promoting the secretion of BDNF (as shown in Figure 4). Notably, PCB-based NPs can increase the recruitment of Aβ into microglia, which can significantly improve Aβ phagocytosis, and when Aβ is degraded, NPs can enter the proteasomal pathway. Numerous major pro-inflammatory cytokines, including IL-1, interferon-γ (IFN-γ), IL-6, and interleukin 17A (IL-17A), were increased in the brains of APP/PS1 mice following administration of the nanomedicine system. The APPswe/PS1dE9 animals demonstrated a reduction in Aβ load, neuronal injury, cognitive impairments, and neuro-inflammation in the brain. Thus, the MCPZFS NPs have excellent potential to function as an “Aβ cleanser”, offering a fresh perspective on therapeutic strategies for AD treatment. Zhang et al. developed a glycosylated “triple-interaction” stabilized polymeric siRNA nanomedicine (Gal-NP@siRNA) that targets BACE1 in an APP/PS1 transgenic AD mice model. Gal-NP@siRNA demonstrated superior blood stability and efficiently crossed the blood–brain barrier (BBB) via a glycemia-controlled glucose transporter-1 (Glut1)-mediated transport, suggesting that siRNAs reduce BACE1 expression and modify metabolic pathways. Remarkably, Gal-NP@siBACE1 reversed the loss in cognitive ability in mice with Alzheimer’s disease without any significant adverse effects. This Trojan horse strategy validates the efficacy of RNA interference therapy for neurodegenerative disorders [130]. See Figure 5 and Figure 6.

There are two immunization strategies, named active and passive immunization-based immunotherapy, for AD [267]. Active immunization is long-lasting, as it induces the immunological memory of the patients [268]. Active vaccines are the most straightforward way to administer vaccines (involving different routes), but their production is not cost-efficient [269]. Immunization produced a polyclonal response; multiple epitopes on the target protein surface with diverse affinity and avidity could be recognized and accessed through antibodies [270]. Alternatively, the immune response also strongly depends on the host immune system; therefore, various patients’ antibody responses may vary from individual to individual [271].

### 6.3. Parkinson’s Disease (PD)

Parkinson’s disease (PD) is the second most prevalent neurodegenerative disorder characterized by motor and non-motor (psychiatric, cognitive, sleep, and autonomic) features, affecting over 10 million people, worldwide [272]. The etiology of PD is mostly attributed to the genetics of individual patients or environmental factors [273]. However, the aggregates of α-synuclein protein (α-syn) form intracellular inclusions known as Lewy bodies, which are considered the neuropathological hallmark of PD [274]. Based on these facts, PD is also called an synucleinopathy and, similarly, includes multiple system atrophy (MSA), pure autonomic failure [41], and Lewy body dementia (LBD) [275]. Nanomedicine-based immunotherapies may offer a novel and alternative strategy for enhancing the immune response against synucleinopathies and Lewy body dementia [276]. Cell-based therapies are the recent research frontier, which has provided an enhanced understanding of the disease, and, possibly, some induced pluripotent stem cell (iPSC) therapies could be used for personalized therapy [277,278].

The technique for gene delivery is also used in regards to PD. The common methods for gene delivery involve vectors of a virus to induce toxicity and immunogenicity. Conversely, an approach using nanotechnology is free from such problems [279]. The complexes composed of polyethyleneimine nanogels and PEG with antisense oligonucleotide efficiently crossed the BBB in vitro, when a functional gel with insulin or Tf molecules was injected into a vein [280]. In another study in a rat model of PD, using 6-hydroxydopamine (6-OHDA) tyrosine hydroxylase, and Tf receptor antibodies combined with PEGylated liposomes reversed impairment after a single administration into the veins. A recent investigation clarified that growth factor for nerves binds with PBCA nanoparticles [281], and L-Dopa-capsulated nanoparticles crossed the BBB to treat prominent basic signs and symptoms of PD [282]. Nanoparticles such as mPEG PLGA with a size of 70 nm, e.g., Schisantherin A (SA), were used to combat Parkinson’s diseases (PD) in the larvae of zebrafish. The SA encapsulated in nanoparticles increased in circulation and uptake by the brain to provide an efficient treatment for PD. SA delivery is more effective than a suspension of SA alone. In conclusion, SA NPs play a neuroprotective role in zebrafish and cellular models of PD.

Recently, PD’s effects on the immune system and microglia have been characterized, which showed that microglia are the major scavenger for extracellular α-syn aggregates, resulting in an increased inflammatory response. These responses were mediated by proinflammatory cytokines including interleukin 1-β (IL 1-β), interleukin-6 (IL-6), interferon γ (INF-γ), and tumor necrosis factor-alpha (TNF-α) [283]. Currently, the gold standard for PD treatment continues to be orally administered dopamine agonists, such as levodopa [284]. Numerous neuro-inflammatory effects, including microgliosis, astrocytosis, and infiltration of T leukocytes, have been shown in the midbrain of PD patients and rat models [285]. A considerably high amount of pro-inflammatory cytokines (TNF-α, IL-1β, and interferon-γ) are released, and oxidative stress and proinflammatory markers such as reactive oxygen species (ROS) and nitric oxide (NO) are produced; in PD, activated astrocytes and microglia are usually linked with BBB impairment [286].

Currently, there is no therapeutic strategy available to treat PD. However, some tentative treatment strategies have been proposed including reducing α-syn expression with either small interfering RNAs (siRNA) or anti-sense RNA and reducing α-syn aggregation using small molecules. In addition, increasing the clearance of α-syn with drugs that promote autophagy prevents the seeding and prion-like spreading of α-syn [287,288]. On the other hand, some immunotherapy studies have shown that vaccination against α-syn reduced α-syn accumulation through activation of autophagy [289] or microglial pathways [290]. Furthermore, specific passive or active immunization strategies such as those with monoclonal antibodies can recognize the epitopes of non-amyloid β component (NAC) and C-terminus of α-syn ameliorated the behavioral deficits and α-syn accumulation in neurons [291] and glial cells [292]. Toll-like receptors (TLRs) initiate innate immune responses through mitogen-activated protein kinase (MAPK) and nuclear factor-kappa B (NF-ҝB) signaling pathways. Microglial cells with strong TLR-induced α-syn stimulate responses, such as TLR2/1- and TLR7-mediated responses, by supplementing the excretion of IL-6 and chemokine immune checkpoint proteins [293].

Recently, several studies used fluorescent quantum dots to inhibit amyloidosis [68]. Graphene quantum dots (GQDs) have shown potential in inhibiting α-synuclein and amyloid pathogenesis [294]; the GQDs efficiently penetrated the BBB due to their small size and antibodies on their surfaces. After a treatment lasting only 7 days, the aggregation and dissembling process has abolished. Moreover, the use of GQDs has a protective role against dopamine neuron loss that is caused by α-synuclein fibril or aggregation formation. The protection of dopaminergic neurons exposed to alpha-synuclein inhibited the formation of fibrils. These GQDs can interact with alpha synuclein fibrils in vitro, to stop their development and even promote their disaggregation. Therefore, if it is considered safe, this kind of nanomedicine could be the magic bullet for PD and other neurogenerative diseases.

## 7. Huntington’s Disease

Progressive motor, cognitive, and psychiatric symptoms describe Huntington’s disease (HD), which is characterized by neuronal loss in the striatum and other brain regions. A mutation in exon 1 of the Huntington gene causes polyglutamine expansion (poly Q), as well as misfolding and accumulation of the Huntington protein (HTT) in the brain [295]. Numerous simulations have characterized the function of astrocytes in HD. Mutant HTT accumulates in striatal astrocytes in the brains of HD patients and HD mouse models, leading to age-dependent HD-like illness and early mortality. There are indicated and preventative treatment options for HD, but none of them are effective enough to entirely cure the disease. Tetra-benazine is currently the only FDA-approved medication for HD treatment [296].

A probable connection between oxidative stress and neurodegenerative disorders, including HD, has been revealed in the literature [297]. Therefore, antioxidant treatment can work in HD to avert oxidative stress. Fullerenes have the capability to clear ROS and decrease oxidative load in cells. Jin et al. described their aggressive behavior on glutamate receptors and, consequently, they can be used for neuroprotective purposes [298]. Nitrendipine is a calcium channel blocker that reduces the incidence of dementia in HD patients by up to 50% over a two-year period. Due to its hydrophilic nature, this medication has absorption issues and, therefore, cannot efficiently cross the BBB. SLNs of nitrendipine were labeled, and a comparison was made between the uptake of bulk medication and nano-formulations. The results reveal that the drug was absorbed more efficiently when encapsulated in SLNs. In addition, short-interfering RNA (siRNA)-encapsulated cyclodextrin nanoparticles that inhibit HTT mRNA expression in vivo and in vitro were investigated [299].

Solid–lipid NPs conjugated with curcumin were used to target mitochondrial imperfections (affecting complex II activity) as a treatment for HD. In the same experiment, 3-nitropropionic acid upregulation of Nrf2 mRNA levels was also established [300]. Another method conjugated rosmarinic acid (RA) with solid–lipid NPs targeting Huntington’s disease, based on RA’s brain-targeting efficacy [301]. Recently, in a *Caenorhabditis elegans* model of HD, Cong et al. used selenium (Se) as a targeted delivery system. Se and selenoproteins were observed to act as neuro-protective compounds and neurodegenerative pathway regulators. Selenium deficiency in the brain confirmed direct correlation with mutant Huntington aggregation, an increase in oxidized glutathione levels, and brain dysfunction [302]. Se is known to reduce oxidative stress in tissues and reduced levels or deficiency of Se is associated with many neurodegenerative disorders, including HD; thus, Se can be considered a therapeutic agent in the future, specifically in a particular formulation. At the present time, many investigations are ongoing to identify new therapeutic targets for the treatment of HD, which is an urgent requirement as Huntington’s disease poses an enigmatic threat to patients.

## 8. Ongoing Clinical Trials and Approved Nanomedicines

The more than 250 US FDA-approved nanodrugs are available on the market is evidence of the success of NPs in clinical trials. There are several interesting drugs for treating multiple myeloma, schizophrenia, lymphomatous meningitis, and MS, including Doxil (doxorubicin HCL liposome injection), Invega Sustenna (paliperidone palmitate), DepoCyt (liposomal cytarabine), and Plegridy (PEGylated interferon B-1a) [303]. Over 33% of the drugs available on the market are in the form of liposomal formulations, which have been identified as the most commonly used nanodrugs [304]. Furthermore, NMs have been studied in a number of clinical trials to establish their relevance in therapeutic settings. In prior research, magnetic nanoparticles and decreased radiation were found to enhance overall survival in glioblastoma patients when compared to traditionally treated peers [305]. Furthermore, nanoparticles play an important role in reducing the toxicity caused by conventional drugs. The delivery capability and distribution of chemotherapeutics (e.g., temozolomide) in intracranial tumor regions in dogs using magnetic NPs should also be mentioned [306]. The most current research confirmed that treating migraine sufferers with curcumin nanoparticles and omega-3 fatty acids considerably lowered inflammation by inhibiting the production of TNF-a, ICAM-1, and COX-2/inducible nitric oxide synthase (Table 2) [307,308,309]. The evidence presented above suggests that NPs are capable of facilitating drug delivery, generating synergistic effects, and reducing the toxicities of drugs in brain diseases by enhancing drug delivery and inducing synergistic effects.

## 9. Conclusions

Recent breakthroughs in medicine, biochemistry, protein engineering, and materials science have led to nanoscale targeting techniques that can transform CNS-based therapeutics. Despite developments in nanotherapeutics, treatments for CNS disorders are not used in clinical practice. Conventional glioblastoma treatment encounters substantial tumor resistance, resulting in poor clinical outcomes. Many researchers are developing new therapeutic techniques to target tumor resistance and improve clinical outcomes. Nanobiotechnology may improve the delivery of immunological therapeutics for CNS-related disorders, such as antibodies, cytotoxic medicines, vaccination antigens, and siRNAs. Nanotechnology has helped immunotherapy reduce tumor or disease progression and improve patient survival. Some nanotechnologies convey synergistic medication combinations and improve innate and adaptive immune responses to fight glioblastoma. These approaches are more effective when using rationally designed nanoparticles with stimuli-sensitive nanoscale materials (i.e., pH and temperature) or inside water-soluble hydrogels and matrices, which can dynamically target immune cells to release the immunotherapeutic once it reaches a certain concentration in the TME. Few studies have examined the role of nano-delivery systems in stimulating innate and adaptive cellular immune responses in glioblastoma patients.

Cancer treatment requires stimulating the immune system without unmanageable side effects. Nanomedicines for glioblastoma immunotherapy require careful design. First, unique nano-immunomodulators are needed to activate macrophages and DCs. Nanocarriers with selective targeting ligands (proteins, peptides, and aptamers) or cell membrane-derived vesicles are designed to actively penetrate barriers and approach APCs. This glioma-specific design differs from standard drug carriers, which are changed (e.g., PEGylation) to avoid phagocytes. Glioblastoma is highly diverse, making it difficult to analyze clinical trial therapy efficacy. Understanding GBM pathogenesis is important when building nanomedicines.

Due to AD’s multiple physiological components and complexity, standard therapy techniques are ineffective. AD has limited treatment techniques despite long-term treatment and research efforts. Nanotechnology offers a helpful alternative to advance research by modulating pathways in specified regions. Specific amyloid antibodies mitigate toxicity and activate clearance pathways. Effective amyloid clearance relies on antibody-mediated or antibody-independent methods that target T cells or microglial cells with amyloid immunotherapy. Indirect activation of microglial innate immune receptors may elicit amyloid clearance without T- or B-cell responses.

Reducing amyloid levels can also entail non-immune strategies, including proteases that degrade toxic amyloid peptides. AD immunotherapy may be paired with nanomedicines or vaccinations. Current nanomedicines or vaccines are hazardous. More biocompatible nanoscale materials are needed to carry AD therapy from lab to bedside. To pass clinical trials, nanotechnology’s pharmacokinetics and toxicology must be improved.

Therapeutic approaches that inhibit α-syn production, toxicity, and aggregation make PD preclinical trials difficult. Synucleinopathies are hallmarks of PD disorders, but their complexity and structural variation make nanomedicine-based immunotherapy problematic. Syn may change conformation. Developing nanomedicine-based immunotherapy is difficult. Due to syn’s structural diversity based on environmental factors or patient variation, more careful designs for immunotherapy are needed. Small nanoparticle-based nanomedicines are reported to decrease PD progression; however, biosafety problems have prevented clinical studies.

## Figures and Tables

**Figure 1 molecules-28-01283-f001:**
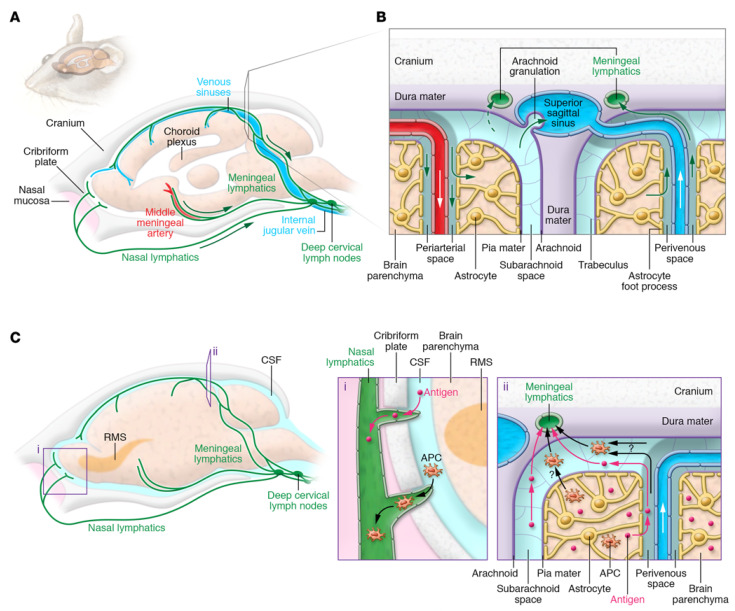
The cervical lymph nodes through the CNS’s lymphatic and lymphatic drainage pathways. (**A**) The meningeal lymphatic vascular system is depicted schematically in the mouse brain. Along with the venous sinuses, arteries, and cranial nerves, the dural lymphatic vessels leave the cranium through the foramina in alignment with the dural blood vessels and cranial nerves. Along with the olfactory nerves, certain lymphatic veins may be seen crisscrossing the cribriform plate. Tracers injected into the SAS or brain parenchyma drain into the dcLNs adjacent to the jugular vein through the dural lymphatic arteries. (**B**) ISF and CSF circulation up close. CSF and solutes are transported into the brain via the perivascular glymphatic drainage system through the periarterial pathway, whereas ISF and solutes are transported out of the brain by the perivenous glymphatic pathway. CSF macromolecules and immune cells are mostly carried by the dural lymphatic channels into the lymph nodes and extracranial systemic circulation, and CSF can reach the venous system via arachnoid granulations. (**C**) Routes for antigens and antigen-presenting cells to exit the central nervous system (APCs). Dendritic cells, in particular, may migrate along the rostral migratory stream (RMS) to enter the lymphatics via the olfactory bulb’s SAS. Alternatively, antigens and APCs are proposed to leave the CNS via the glymphatic pathway (as demonstrated for antigens), reaching the SAS and entering the meningeal lymphatic vasculature via SAS and trafficking to the dcLNs. APCs in the meningeal spaces may also go to the dcLNs through meningeal lymphatic channels. It is still unknown how much each mechanism contributes to cell and antigen outflow. Reproduced with permission from [47].

**Figure 2 molecules-28-01283-f002:**
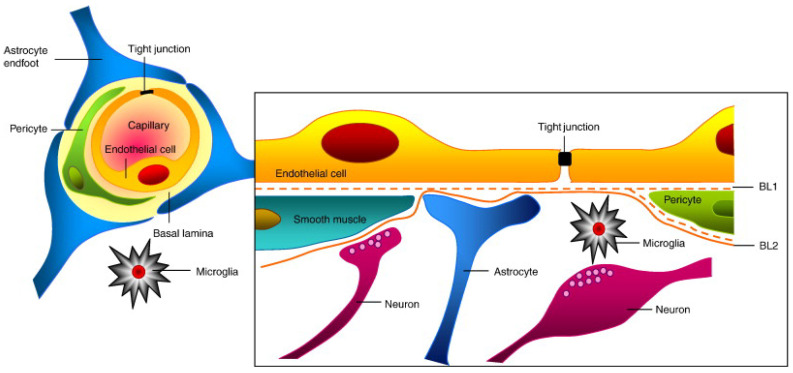
BBB’s structure and composition. The brain endothelium, astrocytes, extracellular matrix, and endothelial cells form tight junctions, which make up the majority of the BBB. Image reproduced with permission from [91].

**Figure 3 molecules-28-01283-f003:**
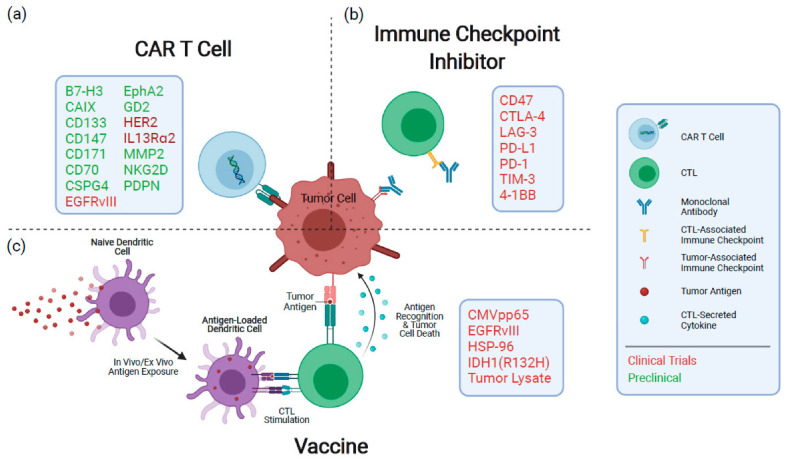
An overview of the immunotherapeutic techniques currently under investigation for the treatment of GBM. (**a**) CAR T cells identify antigens via a genetically designed extracellular receptor that, following antigen binding, induces intracellular T cell activation and degranulation. (**b**) Immunological checkpoint protein inhibitors limit the dampening of immune responses during activation and exhaustion. (**c**) Vaccines expose antigen-presenting cells to tumor antigens, inducing an immune response specific to the target antigens. Therapeutic targets or mediators being pursued for each modality are denoted in the boxes. CAR: chimeric antigen receptor; CTL: cytotoxic T cell. Reproduced with permission from [191].

**Figure 4 molecules-28-01283-f004:**
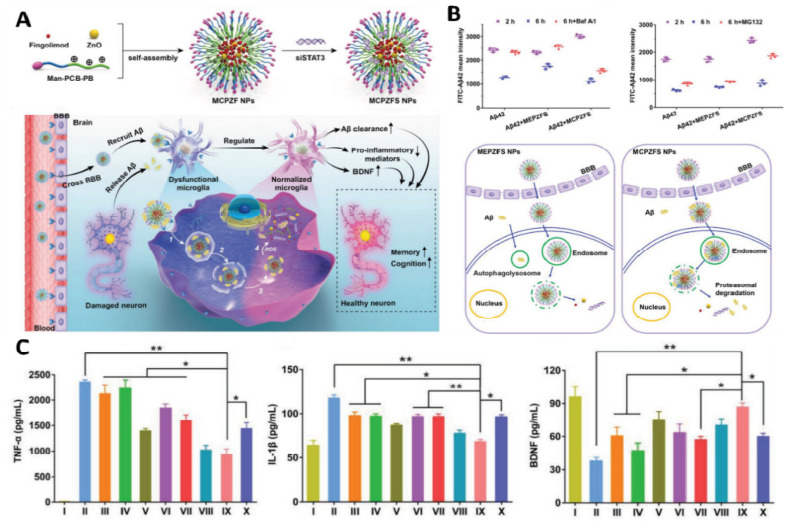
Immune-modulating effect of a PCB-based zwitterionic nanoparticle for AD therapy. (**A**) MCPZFS NPs for AD therapy: structure, preparation, and mechanism. MCPZFS NPs can penetrate the BBB and endocytose into microglia cells to normalized malfunctioning microglia. Pro-inflammatory mediators were reduced. Microglia’s phagocytic capacity was restored when BDNF production rose. The injured neuron is then healed in numerous ways. (1) NPs endocytosed amyloid-β into microglia. (2) After perturbing the endosome/lysosome membrane, NPs leaked into the cytoplasm. (4) Finally, ROS-mediated release of fingolimod, siSTAT3, ZnO, and amyloid-β. (**B**) The effect of NPs on microglial phagocytosis and Aβ degradation after 2 h co-incubation with FITC-A42 and NPs. Flow cytometry was utilized to detect the degradation of A42 of BV2 in BafA1 or MG132 (top right) and the phagocytosis and degradation behavior mediated by MEPZFS NPs and MCPZFS NPs. (**C**) Representing the effect of NPs on the inflammatory regulation of microglia via the ELISA method was applied to determine the levels of TNF-α, IL-1β, and BDNF in the supernatants. Samples: (I) PBS, II) Aβ42, (III) Aβ42 + fingolimod, (IV) Aβ42 + siSTAT3, (V) Aβ42 + CPFS, (VI) Aβ42 + CPZS, (VII) Aβ42 + CPZF-siNC, (VIII) Aβ42 + CPZFS, (IX) Aβ42 + MCPZFS, and (X) Aβ42 + MEPZFS. Copyright, 2019, Wiley and Sons IncReproduced with permission from [265]. * *p* < 0.05, ** *p* < 0.01.

**Figure 5 molecules-28-01283-f005:**
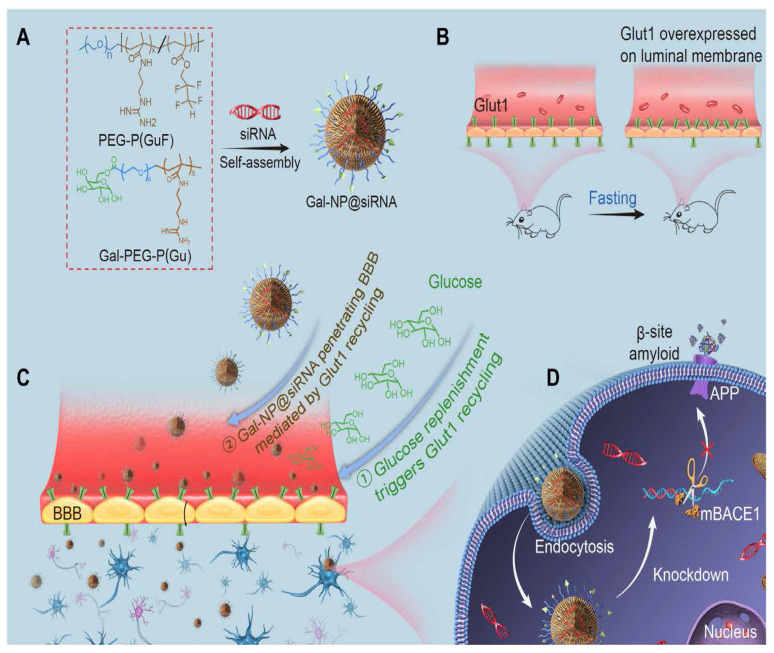
Illustration of glycosylated “triple-interaction” stabilized siRNA nanomedicine (Gal-NP@siRNA) and strategy of treating AD pathology in APP/PS1 transgenic mice. (**A**) Schematic of Gal NP@siRNA manufacturing. (**B**,**C**) How Gal-NP@siRNA enters the brain and accumulates. 24 h fasting increases BBB luminal Glut1 expression. After treatment with Gal-NP@siRNA, glucose replenishment in fasting mice leads to Glut1 recycling from the BBB luminal to the abluminal membrane. (**D**) Gal-NP@siRNA-mediated BACE1 mRNA knockdown reduces amyloid plaques. Adapted with permission from [266].

**Figure 6 molecules-28-01283-f006:**
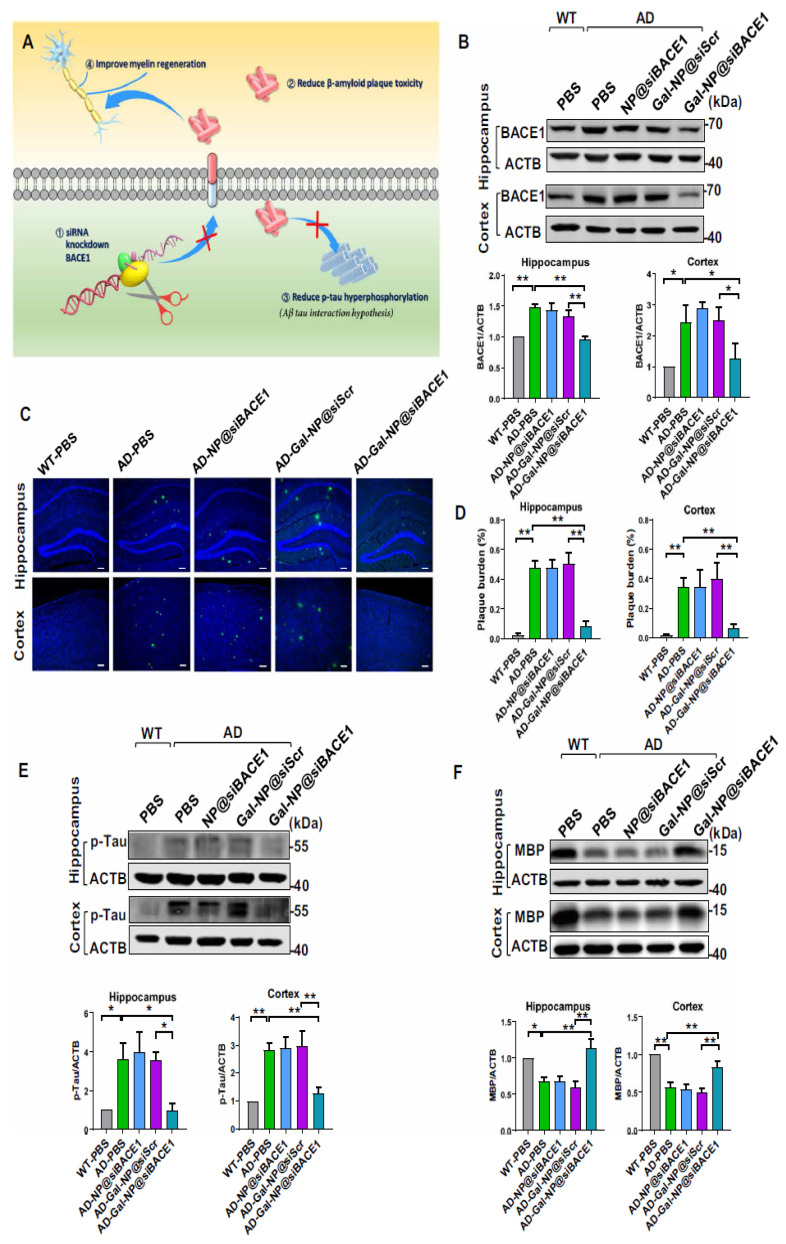
Gal-NP@siBACE1 treatment modulates AD hallmarks in APP/PS1 mice. (**A**) A mechanism for siBACE1′s actions. (**B**) BACE1 protein expression in the hippocampus and cortex of nanocarrier-treated APP/PS1 mice, control APP/PS1 groups, and WT mice. BACE1 expression was quantified relative to actin (n = 3, mean ± SEM, * *p* < 0.05, ** *p* < 0.01). CLSM imaging data to assess amyloid plaque load. A plaque (green) in APP/PS1 transgenic and WT mice hippocampus and cortex. DAPI staining nuclei (blue), 100 µm scale bars. (**C**) Representative confocal laser scanning microscopy (CLSM) imaging data are assessing the amyloid plaque burden. Immunofluorescence of Aβ plaques (green) in the hippocampus and cortex from APP/PS1 transgenic and WT mice. Nuclei were stained by DAPI (blue). Scale bars, 100 µm. (**D**) Amyloid plaques were measured in the hippocampus (**left**) and cortex [181] (n = 4, ** *p* < 0.01; mean ± SEM). (**E**) p-tau and (**F**) MBP expression in the hippocampus and cortex for nanocarrier-treated APP/PS1 mice, control APP/PS1 groups, and WT mice (**top**). Quantification of Western blotting analysis was relative to β-actin (**bottom**) (*n* = 3, mean with SEM, * *p* < 0.05, ** *p* < 0.01). All samples were collected after 10 administrations of nanomedicine. Adapted with permission from [266].

**Table 1 molecules-28-01283-t001:** List of nanomedicines targeting CNS diseases.

Name of Nanomedicine	Disease Name	Purpose	Targeting Plan	Outcomes	Reference(s)
LBNP	GBM	Anticancer, in vivo image	17 peptides	Enhanced bioavailability	[130,131,132]
	PD	PD treatment	Lamp-2b Chlorotoxin	Targeted delivery	[133,134]
	AD	AD treatment	Lamp-2b	Enhances drug delivery, efficiency, and accessibility	[135]
Au-NP	GBM	PDT, PTT	Transferrin peptide RVG29 Peptide	Improves the cellular intake/enhances the efficiency of photodynamic therapy	[136,137,138]
	PD	PD Treatment	chitosan	Enhances the efficiency of the	[139]
	AD	AD treatment	RVG29 Peptide	amyloid-β inhibitor	[140]
PNP	GBM	Anticancer	AS14111 aptamer Transferrin Pep-1 Angiopep	Enhance efficiency and anti-glioma	[141,142,143,144,145]
	PD	PD treatment	ApoE	Enhance the neuroprotective efficiency	[146,147,148,149]
	AD	AD treatment PET	ApoE ^125^I-clioquinol	Beta amyloid-induced cytotoxicity is enhanced by curcumin	[150,151]
IO-NP	GBM	MRI/TEM	Chlorotoxin, chitosan, Anti-EGFRvIII	Targeted therapy and enhanced delivery	[152,153]
	PD	MRI	Anti-ferritin	Detection	[154]
	AD	MRI	Anti-AβPP, Anti-ferritin	Improved amyloid-β revealing	[86,155]

**Table 2 molecules-28-01283-t002:** Treatment trials using NPs for brain diseases and disorders.

NPs	Disease	Mechanisms	Effects	References
Silver NPs	Acute occulsive hydrocephalus	Ventricullitis caused by catheters is prevented	Enhanced the health of patients	[310]
Magnetic iron oxide NPs + reduced radiotherphy	Glioblastomia multiformie	Increased Caspase-3, heat shock protein, and programmed death ligand 1 levels suppress tumor growth	Enhanced the overall survival rate of patients	[305,311]
Ultrasmall magnetic iron oxide	Ischemic stroke	Activates macrophages	Targeted inflammatory cytokines more effectively	[312]
Nano-curcumin + ω-3 fatty acid	Migraine	Intercellular adhesion molecule 1, TNF-a, and cyclooxygenase-2/inducible nitric acid gene expression is suppressed	Relieved headaches	[307,308,309]

## Data Availability

Not Applicable.

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
