# Peer review of "Role of Nanomedicine-Based Therapeutics in the Treatment of CNS Disorders"

_molecules, 2023, doi:10.3390/molecules28031283_

Round 1
Reviewer 1 Report
1. The review is well written and well explained the topic “The Role of Nanomedicine-Based Therapeutics in the Treatment of CNS Problems”; however, boron based drugs/ Nano medicine are totally ignored.
2. All manuscript need to revise to remove grammatical and small error such as, first mention full form and then abbreviation for “BBB”, “BBTB”, etc.,
3. Revised or copy a clear picture of Figure 5, because it’s writing is not clear.
4. Most of the cited references are from 2005-2018. Only few from 2020 or 2021. Please also cite some recent work from 2020 onwards.
Author Response
Reviewer 1
The authors are thankful to the reviewer for understanding the purpose of this manuscript, making comments and giving valuable suggestions to improve the quality of the manuscript.
- The review is well written and well explained the topic “The Roleof Nanomedicine-Based Therapeutics in the Treatment of CNSProblems”; however, boron based drugs/ Nano medicine aretotally ignored.
- Boron-containing compounds (BCCs) have a wide range of medicinal effects[5-7]. Currently, only a few BCCs, such as vaborbactam, tavaborole, crisaborole, and bortezomib, are permitted for use in people. However, many more BCCs are demonstrating promising results as prospective therapies for human disorders, particularly neurological conditions. Consequently, the impact of BCCs on models of neuronal illness is rising. These include studies that reveal multitarget beneficial activity, those that demonstrate a pro-survival impact on human-derived SH-SY5Y cells, such as an A-toxicity model, and those that demonstrate ameliorative abilities in both in vitro and in vivo models of AD[8-11].
- All manuscript need to revise to remove grammatical and small
error such as, first mention full form and then abbreviation for
“BBB”, “BBTB”, etc., - Sorry for the inconvenience. The manuscript has been revised considerably.
- Even with a disrupted blood-brain barrier (BBB), certain invading cancers contain no enhancing areas, which restricts therapeutic medication interaction despite the immune system's dynamic and adaptive character.
- Revised or copy a clear picture of Figure 5, because it’s writing
is not clear.
- Sorry for the inconvenience. The figure has been revised considerably.
- Most of the cited references are from 2005-2018. Only few from
2020 or 2021. Please also cite some recent work from 2020
- Sorry for the inconvenience. The references has been revised considerably and cited 2020 onwards works.
Reviewer 2 Report
The topic chosen by the authors for this contribution is interesting but the reader gets lost due to a too long introduction.
In my opinion, the authors start to really treat the subject only from the paragraph 6.
The previous paragraphs (1 to 5) are often repetitive and difficult to understand. This part is, in my opinion, off-topic with respect to the title of the review and should be re-elaborated in order to give to the reader only the information necessary to understand how Nanomedecines represent an asset to fight against CNS problems.
Figure A is not helping to the comprehension, as well as fig.1.
Table 1: It would be interesting to indicate not only the name of the NP but their nature too.
Author Response
Reviewer 2
The authors are thankful to the reviewer for understanding the purpose of this manuscript, making comments and giving valuable suggestions to improve the quality of the manuscript.
- In my opinion, the authors start to really treat the subject only fromthe paragraph 6. he previous paragraphs (1 to 5) are often repetitive and difficult to understand. This part is, in my opinion, off-topic with respect to the title of the review and should be re-elaborated in order to give to the reader only the information necessary to understand how Nanomedecines represent an asset to fight against CNS problems.
- Sorry for the inconvenience. The introduction section has been revised considerably.
- Nanomedicine is a new field that utilizes nano-scale resources with applications in the diagnosis and therapeutics of disease [1]. Several nano-scale materials are used in nanomedicine: organic, inorganic, carbon-based, polymeric, liposomes, extracellular vesicles, and metals [2-4]. Boron-containing compounds (BCCs) have a wide range of medicinal effects[5-7]. Currently, only a few BCCs, such as vaborbactam, tavaborole, crisaborole, and bortezomib, are permitted for use in people. However, many more BCCs are demonstrating promising results as prospective therapies for human disorders, particularly neurological conditions. Consequently, the impact of BCCs on models of neuronal illness is rising. These include studies that reveal multitarget beneficial activity, those that demonstrate a pro-survival impact on human-derived SH-SY5Y cells, such as an A-toxicity model, and those that demonstrate ameliorative abilities in both in vitro and in vivo models of AD[8-11]. The relationship between nanoscience and pharmaceutical science is broad and there are emerging applications in various fields of disease diagnosis and therapeutics [12-14]. Tumor cells are passively targeted by nanomedicines through enhanced retention and permeability[15]. The improved permeability and retention (EPR) are used to target lymphatic and microvascular systems in the tumor’s interior[16-18]. Alternatively, active and functional targeting nanomedicine approaches are widely examined to overcome particular challenges in tumor theragnostics [2, 19].
- The term "central nervous system (CNS) disorders" refers to a broad range of significant neurological conditions, the majority of which have no effective treatments[20]. Gliomas and glioblastomas, two types of brain cancer, are inherited diseases that originate from cells known as neuroglial ancestor cells[21]. Glioblastoma, which makes up 15% of primary brain tumors and 50% of all gliomas, is the most typical primary CNS tumor in adults. Only temozolomide (TMZ) and the antiangiogenic medicine bevacizumab have been approved by the US Food and Drug Administration (FDA) in recent years for the treatment of gliomas [22]. The median survival rate for people with glioblastoma is less than 2 years despite recent treatment advancements[23]. The more common neurodegenerative disorders Parkinson's disease (PD) and Alzheimer's disease (AD) affect millions of people worldwide [24]. There are several drugs for AD that have been licensed by the FDA, such as cholinesterase inhibitors (such as rivastigmine and donepezil) as well as NMDA receptor antagonists (such as memantine), However, they have limited benefits on severe cognitive impairment and cannot reverse the course of the condition [25-27]. The only drug approved to treat Parkinson's disease is Xadago (safinamide), it reduces motor symptoms without addressing the underlying pathophysiology of the disease [28, 29]. Stopping disease development and treating symptoms and pathology after a late-stage diagnosis are the major challenges in this profession. A new wave of treatment strategies has emerged as a result of these challenges. Currently, radiation, chemotherapy, gene therapy, immunotherapy, and surgery are used to treat CNS illnesses; however, every therapeutic approach has advantages and disadvantages[30, 31]. Targeting disease cells with innate immune responses that are either enhanced or suppressed is how immunotherapy uses the immune system of the host[32]. Active immunotherapy and passive immunotherapy are the two categories of immunotherapeutic techniques[33]. Nanomedicines, tumor vaccines, and non-specific immune stimulants are all examples of active immunotherapeutic techniques that aim to elicit an immune response[34-36]. Through administering immune cells' lymphocytes or antibodies to patients, passive immunotherapy promotes the anticancer effects[37]. Therefore, while implementing immunotherapeutic techniques, it is important to take into account the distinct immunological milieu of the CNS[38, 39]. The presence of complement components, the expression of Toll-like receptors, and the existence of microglia, astrocytes, and pericytes as antigen-presenting cells (APCs) demonstrate that the CNS immune system exists [40]. Even with a disrupted blood-brain barrier (BBB), certain invading cancers contain no enhancing areas, which restricts therapeutic medication interaction despite the immune system's dynamic and adaptive character[41]. Additionally, the distribution of medicines across the BBB is a substantial problem due to the tremendous adaptive characteristics of glioblastoma, moreover to its comparatively low immunogenicity, development of an inhibiting tumor microenvironment (TME), and intertumoral heterogeneity. In fact, many scientists have proposed that the CNS may indeed be regarded as an "immunologically inactive" location, provides a distinct and balanced environment that favors the predominance of immunosuppressive mediator production [42].
- The fundamental challenge that restricts the efficacy of immunotherapies is the engagement of surrogate immunosuppression by brain tumors via several pathways. Overexpressed markers (CD4+, CD25+, and FOXP3+) on regulatory T cells (Tregs) in the sick state influence the immunologically cold TME[43]. Glioblastomas are a particular kind of immunoprivileged tumor since they are sometimes referred to as "cold TMEs" and are only peripherally penetrated by immune cells. A cold TME may also be influenced by immune-suppressive cytokines like interleukin IL-6 and IL-10, as well as immune-suppressive cytokines like transforming growth factor-beta (TGF-β)[44]. PD-1 (programmed cell death protein 1) is overexpressed by inactive Tregs, which aids tumors in evading CNS immune responses and results in a cold TME[45, 46]. As a result, a number of tightly controlled checkpoints maintain brain immunity. Immunotherapy has undergone several changes as a result of training the immune system to detect disease locations[47]. One is immune checkpoint inhibition, in which medications (typically antibodies) block immunological checkpoints that tumors have overexpressed in order to reveal cancer cells and ultimately activate immune responses against the tumors (e.g., melanoma)[48]. Alternately, immune responses can be triggered by genomically altered targeted therapeutics (such as chimeric antigen receptor T cells (CAR T cells)), which have been altered to detect and cure the patient's malignancy[49]. Immunotherapy for the treatment of CNS illnesses continues to be a significant therapeutic issue despite decades of scientific study. For instance, immunological checkpoints and CAR T-cell therapy are less effective in treating nonresponsive cancers that have fewer mutations and neoantigens[50]. Immunotherapy can be particularly difficult due to the location and morphologic similarity of nonmalignant cells with neuroglial cells[51]. A significant barrier to getting a better result for CNS illnesses is the difficulty to deliver therapeutically appropriate dosages to the disease region.
- Drug clearance via the kidneys, drug circulation time in the blood, medication penetration through the blood-brain barrier (BBB), and blood-brain tumor barrier(BBTB) are further obstacles to drug delivery to the central nervous system (CNS)[52]. The payload's ability to enter the CNS is frequently constrained by a natural defense mechanism of efflux pumps, such as multidrug-resistant protein [53] and convection-enhanced diffusion[54]. In order to overcome these difficulties, CNS treatments are being developed using an active and functionally focused nanomedicine-based strategy[55]. Rising field of nanomedicine uses nanoscale materials for a wide range of purposes in diseases diagnosis and treatment[1, 56]. The BBB needs to be crossed by effective nanomedicines for CNS illnesses, and different parameters need to be tuned (e.g., shape, size, functional surface chemistry, circulating half-life, structural stability, permeability, and extravasation)[57]. Additionally, many receptor-facilitated contacts are required to maintain a high degree of surface conjugation chemistry and the related targeting capabilities. One of the primary mechanisms for BBB penetration is receptor-mediated transcytosis (RMT)[58], Transcytosis mediated by adsorption [59] and cell-mediated transport by immune cells, macrophages, and monocytes [60, 61] are further examples. Extracellular vesicles, liposomes, and red blood cell membranes are just a few examples of the various nanoscale materials that have been used as nanomedicines to far[62, 63], additionally to metal nanostructures[64, 65]. Drug-carrying nanostructures significantly enhance the pharmacokinetics and biodistribution of pharmaceuticals in the CNS when compared to free drugs[66, 67].
- Figure A is not helping to the comprehension, as well as fig.1
- Sorry for the inconvenience. The figures are deleted.

Round 2
Reviewer 2 Report
I would like to thank the authors for the corrections they made to the manuscript. I would advise to publish hie corrected version of the rewiew